# Analysis of the Fatty Acid Profile of the Tissues of Hunted Mallard Ducks (*Anas platyrhynchos* L.) from Poland

**DOI:** 10.3390/ani12182394

**Published:** 2022-09-13

**Authors:** Elżbieta Bombik, Katarzyna Pietrzkiewicz, Antoni Bombik

**Affiliations:** Department of Bioengineering and Animal Husbandry, Siedlce University of Natural Sciences and Humanities, 08-110 Siedlce, Poland

**Keywords:** mallard duck, breast muscle, leg muscle, fatty acid profile

## Abstract

**Simple Summary:**

The contemporary human diet contains many times more saturated fatty acids than that of our Palaeolithic ancestors, and the ratio of n-6 to n-3 polyunsaturated fatty acids (PUFAs) is nearly 20 times higher. Lifestyle changes and an unsuitable diet have led to the rapid development of civilization diseases and sudden deaths. This is clearly evident to contemporary consumers, who look for foods produced with respect for the natural environment, which also have an original flavor and health-promoting properties, such as game meat.

**Abstract:**

The aim of the study was to analyse the fatty acid profile of selected tissues of mallard ducks (*Anas platyrhynchos* L.), in relation to where they were obtained and their sex, with regard to the human diet. The study was carried out on material obtained from mallard ducks from two study areas: the Siedlce hunting district and the Leszno hunting district. The research material was the breast and leg muscles of 28 mallards. The samples were frozen and stored at −20 °C. The fatty acid profiles in the biological samples were determined by selected ion recording (SIR). The results showed significantly (*p* < 0.05) lower average levels of saturated fatty acids (SFAs) and monounsaturated fatty acids (MUFAs) and significantly higher (*p* < 0.05) average levels of polyunsaturated fatty acids (PUFAs), n-6 PUFAs, and n-3 PUFAs in the breast muscles of the mallards obtained in the Siedlce hunting district. This in conjunction with the higher (*p* < 0.05) hypocholesterolaemic/hypercholesterolaemic index (h/H) in the leg muscles and lower (*p* < 0.05) atherogenic and thrombogenic indices (AI and TI) in the leg and breast muscles of mallards in the Siedlce hunting district indicate the higher health-promoting value of the meat of ducks from this region. The average n-6/n-3 PUFA ratio in the breast muscles was significantly (*p* < 0.05) higher in mallards obtained in the Leszno hunting district. Males of the species had a significantly (*p* < 0.05) higher average n-6/n-3 PUFA ratio in the breast muscles than females. The PUFA/SFA ratio was significantly (*p* < 0.05) higher in the leg muscles of the female mallards than in the males.

## 1. Introduction

The mallard duck (*Anas platyrhynchos* L.) is a game species hunted in Poland for its lean, tasty meat. Its nutritional, culinary, and processing value derives from the chemical composition of the muscle tissue, connective tissue, and fat in the meat. The high nutritional value of mallard meat is linked to the ducks’ wild origin and behaviour. The innate activity of these birds results in high content of haem pigments and a low-fat content. The carcass contains on average 73% muscle and 2.3% adipose tissue. Mallards have lower fat content and higher protein content than domesticated livestock [1]. According to the Regulation of the Minister of the Environment of 11 March 2005 on establishment of the list of game species [2], the hunting period for the species is from 15 August to 21 December. Hunters are authorized by law to manage game populations [3]. According to estimates [4,5,6], the average number of mallard ducks culled annually in recent years is 75,000. Therefore per capita consumption of the meat of this species in Poland is low. It is not generally available in retail chains. Mallard meat is eaten in Poland mainly by hunters, their families, and people from their immediate vicinity.

The diet of mallards depends on the time of year and periodic availability of a given food. Its composition influences the fatty acid profile of the meat, including the content of unsaturated fatty acids. The food of these birds consists of cereal grains, aquatic plants, aquatic invertebrates, and insect larvae. Hitchcock et al. [7] analysed the diet of wild ducks, including mallards, and found both plant and animal material. English et al. [8] reported that coastal mallards seemed to rely less on invertebrates and more on plant material. Miller [9] also found that wild ducks preferred plant foods to animal foods.

The development of civilization and continually accelerating pace of life have led to interest among consumers in easily accessible food. The contemporary human diet contains many times more saturated fatty acids than that of our Palaeolithic ancestors, and the ratio of n-6 to n-3 polyunsaturated fatty acids (PUFAs) is nearly 20 times higher. According to [10], the n-6/n-3 PUFA ratio in the diet should range from 2:1 to 3:1 and should not exceed 10:1, whereas [11] report that the n-6/n-3 PUFA ratio in the human diet is currently 20:1 or higher. These differences are due in part to the common use of fats rich in n-6 PUFAs, such as maize, sunflower, and soybean oil, and to the availability of meat from livestock animals fed diets rich in these acids.

Lifestyle changes and an unsuitable diet have led to the rapid development of civilization diseases and an increase in sudden deaths. This is clearly evident to contemporary consumers, who look for foods produced with respect for the natural environment, which also have an original flavour and health-promoting properties, such as game meat.

Saturated fatty acids (SFAs) can be synthesized in the human body, in which they mainly serve as a source of energy, and therefore there is no need for additional supplementation. SFAs are present in all dietary fats, but their main source in the diet is animal products.

The presence of double bonds influences the biological value of fat. Among the four families of unsaturated fatty acids, biological activity is exhibited mainly by the n-3 family, whose precursor is α-linolenic acid (ALA; C18:3n3), and the n-6 family, whose precursor is linoleic acid (LA; C18:2n6c). The essential unsaturated fatty acids (EFA), C18:3n3 and C18:2n6c, cannot be synthesized in the human body and must be supplied with food [12]. The docosahexaenoic acid (C22:6n3) is quantitatively the most important n-3 fatty acid in the cell membranes and plasma membranes of the entire body [13]. The arachidonic and eicosapentaenoic acids (C20:4n6 and C20:5n3) are substrates for the synthesis of eicosanoids [14]. The effect of polyunsaturated fatty acids on the body is largely determined by the effects of eicosanoid activity. Eicosanoids regulate the body’s immune response and inflammatory response, proliferation of tumour cells, cardiovascular function, blood pressure, maintenance of Na^+^ homeostasis, transport of Ca^+^ ions, the level of triglycerides in the blood, clot formation, airway patency, and blood flow in the kidneys [15,16].

The TI and AI (thrombogenic and atherogenic indices) are considered to be better indicators of fat quality than the PUFA/SFA ratio. The lower their value, the more beneficial the fatty acid profile is for human health. This is because not all SFAs have hypercholesterolaemic effects [17].

The available literature contains little information on the fatty acids of the meat of mallard ducks, because the meat of this species is very difficult to acquire and thus not very popular. However, due to its nutritional value, mallard meat could be an attractive culinary resource and an element of regional promotion. The aim of the study was to analyse the fatty acid profile of selected tissues of mallard ducks (*Anas platyrhynchos* L.) in relation to where they were obtained and their sex, with regard to the human diet.

## 2. Materials and Methods

### 2.1. Animals and Sample Collection

The study was carried out on material from mallard ducks reared in two study areas: the Siedlce hunting district, located in the Masovian Voivodeship, and the Leszno hunting district in the Greater Poland Voivodeship (Figure 1). In both the Voivodeships, cereals (wheat, barley, and maize) are dominant in the crop structure [18]. Therefore, it is likely that grains of cereals made up a large percentage of the mallards’ diet. The two regions were distinguished by different environmental resources and different degrees of intensification of agriculture. In the Greater Poland Voivodeship, there are many large farms with intensive crop production, including high levels of mineral fertilizers. Farms in the Masovian Voivodeship have a smaller area and are managed more sustainably, so their environmental impact is smaller [19].

The research material was obtained by hunters who are authorized by law to manage populations of game animals [3]. Mallards were culled in accordance with the Annual Hunting Plan, which included harvesting of wild ducks for the hunting districts analysed in the study, drawn up for the period from 1 April to 31 March of the following year, i.e., for the 2018/2019 hunting season. Ducks were hunted by individuals and groups, always with hunting dogs trained for this type of hunting. This practice guaranteed that all birds shot down would be found, and thus was ethical according to the rules and regulations in force in the Polish Hunting Association. Each person hunting mallards had a permit issued by the head of the relevant Hunting Club, which contained information on the number of individuals the hunter may harvest. Ducks were hunted in three ways: evening blind hunting for birds flying in to feed, morning blind hunting for birds returning from feeding, and stalking during the day in the birds’ preferred places, always in compliance with safety principles. Acquisition of mallards was limited to the first two months of the hunting period for the species, i.e., from 15 August to 15 October, before the birds had begun their migration. The carcasses obtained by the hunters were dissected. The research material was the breast and leg muscles of 28 mallards, including 6 females and 6 males in the Siedlce hunting district and 8 females and 8 males in the Leszno hunting district. The meat samples were frozen and stored at −20 °C.

The composition of the mallards’ diet was probably similar in both hunting districts. It consisted of both animal and plant material, including cereal grains, which could be seen in their stomachs during dissection. It may also have included aquatic plants, aquatic invertebrates, gastropods, and insect larvae. Variation in the composition of the diet of mallards between hunting districts may have been linked to the level of intensification of agriculture.

### 2.2. Laboratory Analysis

The fatty acid profile in the biological samples was determined by selected ion recording (SIR). Fatty acid methyl esters (FAME) in the lipids were analysed by gas chromatography—mass spectrometry (GCMS) in a CLARUS 600 mass spectrometer (Perkin Elmer), using a COL-ELITE-5MS column (60 m × 0.25 mm × 0.25 µm). Based on the fatty acid composition, the following parameters were calculated: sum of SFAs, monounsaturated fatty acids (MUFAs), PUFAs, n-6 PUFAs, and n-3 PUFAs, n-6/n-3 PUFA ratio, PUFA/SFA ratio, hypocholesterolaemic/hypercholesterolaemic index (h/H), atherogenic index (AI), and thrombogenic index (TI).

### 2.3. Statistics

Statistical analysis of the results was performed using Statistica 13.0 software. The numerical data were characterized by means of arithmetic mean ($\bar{x}$), extreme values (min., max.), standard deviation (s), and coefficient of variation (V%). In addition, two-way non-orthogonal analysis of variance (Fisher-Snedecor F test) was performed. A detailed comparison of means was performed using the Tukey test for a significance level of *p* < 0.05. Correlations between features were determined using Pearson’s correlation coefficient (r), and the significance of these relationships was verified for *p* < 0.05 and *p* < 0.01.

## 3. Results

### 3.1. Fatty Acid Profile of Breast Muscles (Anas platyrhynchos L.)

The content of the following fatty acids was significantly higher in the breast muscles of mallards from the Siedlce hunting district in comparison with the Leszno hunting district: C12:0 lauric acid (0.22%; a nearly 4-fold difference), C14:0 myristic acid (0.73%), C15:0 pentadecanoic acid (0.35%), C17:0 heptadecanoic acid (1.04%; more than 2-fold), C18:0 stearic acid (13.06%), C18:1n9t elaidic acid (3.60%), C18:2n6c linoleic acid (20.94%), C18:3n3 α-linolenic acid (0.73%), C20:4n6 arachidonic acid (4.51%), C20:5n3 eicosapentaenoic acid, and C22:6n3 docosahexaenoic acid (nearly 2-fold) (Table 1).

The content of the following fatty acids was significantly higher in the breast muscles of ducks from the Leszno hunting district in comparison with the Siedlce hunting district: C15:1 cis-10-pentadecanoic acid (1.51%; 2-fold difference), C16:0 palmitic acid (17.47%), C17:1 cis-10-heptadecanoic acid (0.51%; more than 2-fold), C18:1n9c oleic acid (32.16%), C18:3n6 γ-linolenic acid (0.20%), C24:0 lignoceric acid (0.09%), and C24:1 nervonic acid (0.17%). The significantly higher content of n-3 unsaturated fatty acids in the breast muscles of mallards obtained in the Siedlce hunting district in comparison with the Leszno hunting district may indicate the beneficial influence of the diet of birds from this region on the fatty acid profile. The analysis also showed differences between the sexes in the content of fatty acids in the breast muscles of mallards. Statistically significantly higher content of the following fatty acids was obtained in the breast muscles of female mallards: C15:0 pentadecanoic acid, C17:0 heptadecanoic acid, C20:3n3 cis-11.14.17-eicosatrienoic acid, C20:3n6 dihomo-γ-linolenic acid, C20:5n3 eicosapentaenoic acid, C22:6n3 docosahexaenoic acid, and C24:0 lignoceric acid (*p* < 0.05). For C20:0 arachidic, C20:1 eicosenoic, and C20:2 eicosadienoic acids, these values were twice as high in females as in males (*p* < 0.05). Fatty acids with significantly higher content in the muscles of males were C12:0 lauric, C16:0 palmitic, C17:1 cis-10-heptadecanoic, C18:0 stearic, C18:1n9t elaidic, C20:4n6 arachidonic, and C22:0 docosanoic acids (*p* < 0.05). The fatty acid profile of the breast muscles of females was more favourable than that of the breast muscles of males.

The level of polyunsaturated fatty acids—PUFAs (C18:2n6c + C18:3n3 + C18:3n6 + C20:2 + C20:3n3 + C20:3n6 + C20:4n6 + C20:5n3 + C22:6n3) (29.945%), n-6 PUFAs (C18:2n6c + C18:3n6 + C20:3n6 + C20:4n6) (26.11%), and n-3 PUFAs (3.34%), as well as the PUFA/SFA ratio (0.97), were significantly higher in the breast muscles of mallards obtained in the Siedlce hunting district in comparison with the Leszno hunting district (Table 2). The content of saturated fatty acids—SFAs (C12:0 + C14:0 + C15:0 + C16:0 + C17:0 + C18:0 + C20:0 + C22:0 + C24:0) (31.68%); monounsaturated fatty acids—MUFAs (C14:1 + C15:1 + C16:1 + C17:1 + C18:1 n9c + C18:1n9t + C20:1 + C24:1) (42.37%); and the n-6/n-3 PUFA ratio (9.82) were significantly higher in ducks from the Leszno hunting district in comparison with the Siedlce hunting district. The significantly higher average level of PUFAs, n-6 PUFAs, n-3 PUFAs, and the PUFA/SFA ratio in the breast muscles of mallards obtained in the Siedlce hunting district in comparison with the Leszno hunting district may indicate richer environmental resources and their effect on the composition of the diet of birds from this region. The breast muscles of male mallards were shown to have significantly higher mean content of SFAs and PUFAs and a higher n-6/n-3 ratio than those of females (*p* < 0.05); whereas the mean content of n-3 PUFAs (C18:3n3 + C20:3n3 + C20:5n3 + C22:6n3) and the PUFA/SFA ratio were significantly higher in the breast muscles of females than in males (*p* < 0.05), indicating better health-promoting properties.

Sex was shown to influence the fatty acid indices of the breast muscles of mallards. The hypocholesterolaemic/hypercholesterolaemic (h/H) index was statistically significantly higher in females (3.56) than in males (3.44), whereas the reverse was noted for the atherogenic (AI) and thrombogenic (TI) indices. The fatty acid indices of the breast muscles were influenced by the location where the ducks were obtained. The h/H index in the breast muscles of mallards from the Siedlce hunting district (3.81) was statistically significantly higher than in the Leszno hunting district (3.26); whereas the AI and TI were statistically significantly higher in the Leszno hunting district (0.29 and 0.87, respectively) than in the Siedlce hunting district (0.26 and 0.82, respectively).

### 3.2. Fatty Acid Profile of the Leg Muscles of Mallard Ducks (Anas platyrhynchos L.)

The leg muscles of birds obtained in the Siedlce hunting district had significantly higher content of the following fatty acids in comparison with the Leszno hunting district: C12:0 lauric (0.32%; more than 3-fold), C14:0 myristic (1.03%), C15:0 pentadecanoic (0.47%), C17:0 heptadecanoic (0.84%), C18:1n9t elaidic (3.18%), C18:3n3 α-linolenic (0.599%), C18:2n6c linoleic (17.120%), C20:4n6 arachidonic (2.51%), C20:5n3 eicosapentaenoic (0.35%), and C20:6n3 docosahexaenoic (0.91%; nearly 2-fold) (Table 3). The significantly higher content of n-3 fatty acids in the leg muscles of mallards obtained in the Siedlce hunting district in comparison with the Leszno hunting district may indicate that their diet was richer in these polyunsaturated fatty acids. The content of the following fatty acids was significantly higher in the leg muscles of ducks from the Leszno hunting district in comparison with the Siedlce hunting district: C15:1 cis-10-pentadecanoic (1.15%; nearly 3-fold), C16:0 palmitic (16.96%), C18:0 stearic (11.09%), C18:3n6 γ-linolenic (0.30%), C20:0 arachidic (0.77%), C20:1 eicosenoic (1.11%), C20:2 eicosadienoic (1.12%), C20:3n3 cis-11.14.17-eicosatrienoic (0.47%), C20:3n6 dihomo-γ-linolenic (0.78%), C22:0 docosanoic (0.28%), C24:0 lignoceric (0.15%), and C24:1 nervonic (0.19%; 2-fold). The sex of the birds was shown to affect the fatty acid profile of the leg muscles. The leg muscles of males had significantly higher average content of C14:0 myristic and C18:0 stearic acid than those of females (*p* < 0.05). The leg muscles of female mallards had significantly higher content of the following fatty acids in comparison with males: C14:1 myristoleic, C15:1 cis-10-pentadecanoic, C16:0 palmitic, C16:1 palmitoleic, cis-11.14.17-eicosatrienoic, and C20:5n3 eicosapentaenoic (*p* < 0.05). The leg muscles of females were shown to have a more favourable fatty acid profile than those of males.

The analyses showed that the levels of PUFAs (23.67%), n-6 PUFAs (20.55%), n-3 PUFAs (2.25%), and the PUFA/SFA ratio (0.78) were significantly higher in the leg muscles of ducks obtained in the Siedlce hunting district in comparison with the Leszno hunting district (Table 4). The content of SFAs (31.256%) and MUFAs (46.91%) and the n-6/n-3 PUFA ratio (10.26) were significantly higher in the leg muscles of ducks obtained in the Leszno hunting district than in birds from the Siedlce hunting district. The significantly higher level of PUFAs, n-6 PUFAs, n-3 PUFAs, and the PUFA/SFA ratio in the leg muscles of mallards from the Siedlce hunting district in comparison with the Leszno hunting district may indicate richer environmental resources in that region and their effect on the composition of the birds’ diet. The analyses showed differences between the sexes in the content of fatty acids in the leg muscles of mallards. The mean content of SFAs and the n-6/n-3 PUFA ratio were significantly higher in the leg muscles of males (*p* < 0.05), whereas the mean content of n-3 PUFAs and the PUFA/SFA ratio were significantly higher in the leg muscles of females (*p* < 0.05), which is indicative of higher health-promoting value.

The sex of mallards was shown to influence the TI index; the males had a significantly higher TI (0.83) than the females (0.82). The differences for the other indices were statistically non-significant. The location where the mallards were obtained was shown to influence the fatty acid indices. The h/H index in the leg muscles of the birds from the Siedlce hunting district (3.45) was statistically significantly higher than in the Leszno hunting district (3.21); whereas the AI and TI indices were statistically significantly higher in birds from the Leszno hunting district (0.30 and 0.84, respectively) in comparison with the Siedlce hunting district (0.29 and 0.80, respectively).

## 4. Discussion

Differences in the diet of the mallards from the two hunting districts may have significantly influenced the fatty acid profile of the muscles of these birds. The nutritional value of the ducks’ diet in a given region may have been affected by differences in environmental resources and in the degree of impact of agricultural production on the environment. Many authors have reported a positive effect of the inclusion of various cereal species in the diet on the fatty acid profile of various animal species [20,21,22,23].

If the proportion of insect larvae in the diet of mallards is too high, it may affect the fatty acid profile in their muscles. Daszkiewicz et al. [24] reported that ≥50% inclusion of full-fat black soldier fly (*Hermetia illucens*) larva meal as an alternative protein source in the diet of broiler chickens is too high, as it negatively affects the fatty acid profile of the meat. Gastropods in the diet of mallards may also affect the fatty acid profile of the muscles. Özogul et al. [25] and Milinsk et al. [26] showed a high proportion of unsaturated fatty acids in the total fat fraction of the meat of gastropods.

In all groups of mallards, the SFA level was higher in the breast muscles than in the leg muscles, which is confirmed by [27]. The SFA level in the breast muscles of mallards ranged from 30.05% to 33.59%. A higher average SFA level in the breast muscles of mallards than in the present study was obtained by [28], and a lower level by [27]. A higher average SFA level in the breast muscles of other poultry breeds and species than the level obtained in the present study was observed in broiler chickens and in Cherry Berry ducks [29,30], as well as in Pekin ducks (AP54, PP54, PP45 and Star H.Y.) [31], which given recommendations to reduce SFAs in the human diet may indicate that the meat of wild mallards is of higher nutritional value. The SFA level in the leg muscles of mallards ranged from 29.94% to 31.73%. A lower average concentration of SFAs in the leg muscles of other breeds and species has been observed by [32] in Pekin and Muscovy ducks and by [33] in Korean native ducks, whereas higher concentrations were reported by [31] in Pekin Star 53 H.Y., AP54, PP54, and PP45 ducks and by [34] in ring-necked pheasants. Wołoszyn et al. [35] showed a higher average level of SFAs in the leg muscles of Pekin P33 and A3 ducks and Miniduck K2 ducks than that observed in the present study in male mallards from both hunting districts. Kijanko et al. [36] reported a higher average SFA level in the liver of mallards (from 43.05 to 47.03%) than that observed in the present study in ducks obtained in both hunting districts.

Organs, including the liver, contain lower levels of MUFAs and higher levels of PUFAs than the skeletal muscles [37]. The breast muscles and legs of the groups of mallards had a higher level of MUFAs than of SFAs, which is beneficial for the consumer. A higher average level of MUFAs was shown in the leg muscles than in the breast muscles, which is confirmed by [27]. The MUFA level in the breast muscles of mallards ranged from 37.16% to 43.46%. A lower average MUFA level in the breast muscles of mallards (23.7%) was reported by [27]. A lower average MUFA level in the breast muscles of mallards than that recorded in the present study was also noted by [28,38]. Compared to the mallards from the Siedlce hunting district, a higher average MUFA level in the breast muscles was found by [39] in White Koluda geese, by [32] in Muscovy ducks, and by [38] in Green-legged partridge and Rhode Island Red capons, turkeys, and chicken broilers. Ali et al. [29] reported a lower average MUFA level in the breast muscles of broiler chickens and Cherry Berry ducks than in the mallards from the Leszno hunting district. The MUFA concentration in the leg muscles of mallards ranged from 45.00% to 48.32%. A lower average MUFA level in the leg muscles compared to that obtained in the present study in mallards was observed in Pekin and Muscovy ducks [32], Pekin Star 53 H.Y., AP54, and PP54 ducks [31], Korean native ducks [33], and farmed ring-necked pheasants [34], which may indicate that the leg muscles of mallards have higher nutritional value.

The average level of PUFAs was higher in the breast muscles of mallards than in the leg muscles, which is consistent with results obtained by [27]. The PUFA concentration in the breast muscles of mallards ranged from 25.68% to 31.74%. [27,28,38] noted a higher average PUFA level in the breast muscles of mallards than in the present study. In other breeds and species of poultry [32,33,39,40], a lower PUFA level was shown in these muscles, which may have been due to differences in the birds’ diet as well as to genotypic differences. The PUFA concentration in the leg muscles of mallards ranged from 20.43% to 23.91%. A higher average PUFA concentration in the leg muscles compared to the mallards from both hunting districts was observed in Pekin and Muscovy ducks [32] and in Korean native ducks [33], whereas a lower level was observed in White Koluda geese [39] and in ring-necked pheasants [34]. Another study showed a higher average level of PUFAs in the leg muscles of male ducks than in the male mallards in the present study [35].

In the present study, all groups of mallards had a higher average level of n-6 PUFAs in the breast muscles than in the leg muscles. The level of n-6 PUFAs in the breast muscles of mallards ranged from 22.359% to 27.685%, whereas in the leg muscles it ranged from 18.09% to 20.75%. A higher average level of n-6 PUFAs in the breast muscles of mallards of both sexes (33.06% and 32.97% in males and females, respectively) was reported by [28]. The present study showed a higher average level of n-6 PUFAs in the breast muscles of mallards obtained in the Siedlce and Leszno hunting districts than that observed in Pekin and Muscovy ducks [32]. Korean native ducks had a higher average level of n-6 PUFAs in the breast and leg muscles than that shown in mallards [33].

In the present study, all groups of mallards had a higher average level of n-3 PUFAs in the breast muscles than in the leg muscles. The n-3 PUFA concentration ranged from 1.88% to 3.56% in the breast muscles of mallards from 1.51% to 2.40% in the leg muscles. In both the present study and in [28], a higher average concentration of n-3 PUFAs was shown in the breast muscles of female mallards than in males. The average level of n-3 PUFAs shown in the breast muscles of male mallards in the present study was higher than that reported by [41] in all groups of birds tested. A higher share of n-3 PUFAs in the breast and thigh muscles of Pekin ducks than that shown in the present study in the mallards from both hunting districts was observed by [32].

The n-6/n-3 PUFA ratio in the breast muscles of mallards ranged from 7.76 to 12.47. For human health, the value of this ratio should be as low as possible. A lower average n-6/n-3 PUFA ratio in the breast muscles of male and female mallards (3.12% and 3.10%, respectively) than that observed in the mallards of both sexes in the present study was reported by [28], which may indicate a higher proportion of foods containing greater amounts of n-6 PUFAs in the diet of these birds. Pekin A44 ducks had a lower n-6/n-3 PUFA ratio than was shown in the present study in mallards from both hunting districts [42]. Wołoszyn et al. [35] observed a lower average n-6/n-3 PUFA ratio in the breast muscles of male Pekin P33 and Miniduck K2 ducks. The results of the present study showed a higher average n-6/n-3 PUFA ratio in the muscles of mallards than that shown in Pekin and Muscovy ducks [32] and in the breast muscles of broiler chickens [30]. Differences in the average n-6/n-3 PUFA ratio between farmed breeds and species of poultry and mallards may have been due to dietary differences, i.e., feed rich in n-3 fatty acids given to farmed poultry. A higher average value for this ratio in the breast muscles and liver of chickens receiving a sunflower oil supplement was reported by [43]. Mallards obtained in the Leszno hunting district had a significantly higher n-6/n-3 PUFA ratio in the leg muscles than birds from the Siedlce hunting district. The values were 10.225 and 9.168, respectively, which are nutritionally favourable. The n6/n3 PUFA ratio in meat is mainly regulated by diet [44]. The significantly higher average n-6/n-3 ratio in the meat of mallards from the Leszno hunting district may have been due to the high intensity of crop production, including high levels of mineral fertilizers. Male mallards had a significantly higher average value for this ratio (10.16) than females (9.42). The n-6/n-3 PUFA ratio in the leg muscles of mallards ranged from 8.64 to 11.96. Wołoszyn et al. [35] reported a lower average n-6/n-3 PUFA ratio in the leg muscles of male ducks compared to the male mallards in the present study.

In all groups of mallards, the average PUFA/SFA ratio was higher in the breast muscles than in the leg muscles. The PUFA/SFA ratio should be at least 0.40 [45]. The ratio in the breast muscles of mallards exceeded this value, ranging from 0.75 to 1.02. The present study showed a lower PUFA/SFA ratio in the breast muscles of female and male mallards obtained in both hunting districts than that reported by [28] for both sexes. Compared to the mallards from both hunting districts, a lower, less favourable PUFA/SFA ratio was found in the breast muscles of chicken broilers and Cherry Berry ducks by [29], in ring-necked pheasants by [34], and in Cherry Valley and Spent Layer ducks by [30]. A highly favourable PUFA/SFA ratio was obtained in the present study in the leg muscles of mallards, ranging from 0.70 to 0.78. Kokoszyński [31] showed a lower average PUFA/SFA ratio in the leg muscles of mallards than in the female and male mallards in the analysed hunting districts. Wołoszyn et al. [35], on the other hand, showed a higher average value for this ratio in the leg muscles of male ducks from four conserved flocks (Miniduck K2, Polish Pekin P33, Pekin population-type A3, and synthetic Polish flock SB) and two breeding strains (A55 and P66).

A relatively high h/H ratio and low AI and TI values reduce the incidence of cardiovascular disease [46]. The higher h/H ratio in the leg muscles and lower AI and TI indices in the leg and breast muscles of mallards from the Siedlce hunting district in comparison to the Leszno hunting district indicate that the meat of ducks from the Siedlce hunting district has higher health-promoting value. The h/H, AI, and TI indices can be indicators of the influence of the fatty acid profile on cardiovascular disease [17]. The more favourable fatty acid profile, high h/H ratio, and low AI and TI in the meat of mallards from the Siedlce hunting district may have been linked to the diet of birds obtained in a region with sustainable management.

The limitations of this study include drought in the hunting districts during the study period and the associated difficulties with obtaining enough mallards to allow for reliable inferences. Therefore, there is a need for further study to confirm the research hypotheses.

## 5. Conclusions

The meat of mallards is of high nutritional value. This is evidenced by the high average proportion of essential fatty acids and low proportion of saturated fatty acids in the breast muscles and the higher proportion of MUFAs than of SFAs in the breast and leg muscles of ducks from the analysed hunting districts. Given that it is recommended to limit the amount of SFAs in the human diet, this is beneficial for consumer health. The significantly higher average concentrations of EFAs, linoleic acid, and α-linolenic acid in the breast and leg muscles and the significantly lower n-6/n-3 ratio in the breast muscles of mallards from the Siedlce hunting district compared to the Leszno hunting district may have been influenced by the diet of the ducks in the Siedlce district, which improved the fatty acid profile of the meat.

## Figures and Tables

**Figure 1 animals-12-02394-f001:**
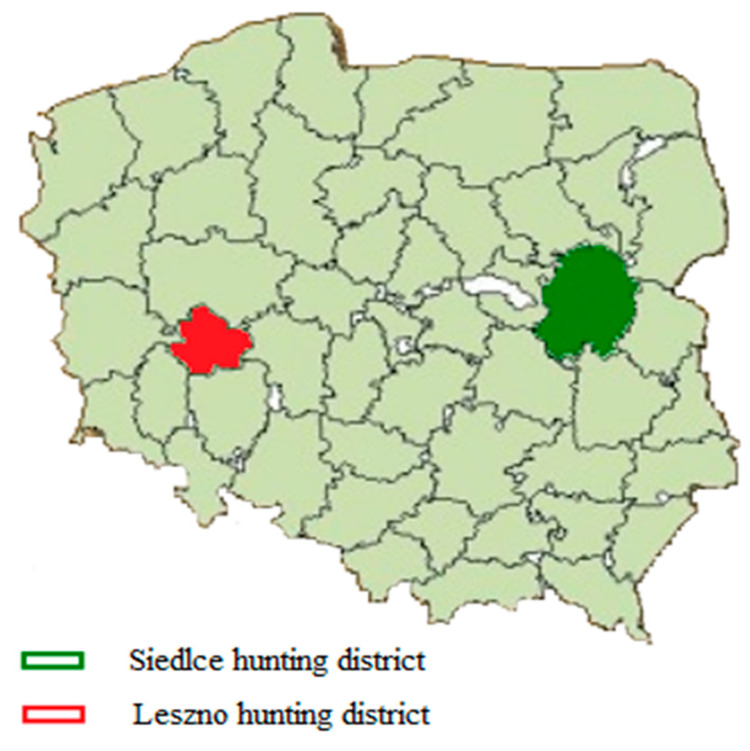
Location of research areas in Poland.

**Table 1 animals-12-02394-t001:** Fatty acid profile of the breast muscles of mallards (*Anas platyrhynchos* L.) (%).

Item	Sex		*p*-Value	Hunting District		*p*-Value
	Female (n = 14) $\bar{x}$ ± s	Male (n = 14) $\bar{x}$ ± s		Siedlce (n = 12) $\bar{x}$ ± s	Leszno (n = 16) $\bar{x}$ ± s	
C_12:0_	0.10 ± 0.09 b	0.14 ± 0.12 a	0.026	0.22 ± 0.07 a	0.04 ± 0.01 b	0.000
C_14:0_	0.69 ± 0.10 a	0.59 ± 0.24 a	0.210	0.73 ± 0.18 a	0.58 ± 0.17 b	0.005
C_14:1_	0.22 ± 0.06 a	0.21 ± 0.07 a	0.881	0.21 ± 0.08 a	0.21 ± 0.05 a	0.866
C_15:0_	0.30 ± 0.04 a	0.23 ± 0.14 b	0.005	0.35 ± 0.05 a	0.20 ± 0.10 b	0.000
C_15:1_	1.40 ± 0.56 a	0.94 ± 0.81 a	0.082	0.72 ± 0.07 b	1.51 ± 0.82 a	0.002
C_16:0_	15.69 ± 0.99 b	16.85 ± 1.72 a	0.004	14.99 ± 0.33 b	17.47 ± 1.03 a	0.000
C_16:1_	4.26 ± 0.48 a	4.03 ± 0.66 a	0.528	4.21 ± 0.69 a	4.09 ± 0.49 a	0.495
C_17:0_	0.82 ± 0.19 a	0.63 ± 0.39 b	0.001	1.04 ± 0.01 a	0.49 ± 0.22 b	0.000
C_17:1_	0.32 ± 0.10 b	0.44 ± 0.24 a	0.006	0.21 ± 0.08 b	0.51 ± 0.15 a	0.000
C_18:0_	12.29 ± 0.97 b	12.91 ± 0.44 a	0.005	13.06 ± 0.59 a	12.26 ± 0.79 b	0.000
C_18:1n9c_	30.76 ± 1.73 a	31.29 ± 1.75 a	0.187	29.51 ± 1.20 b	32.16 ± 1.16 a	0.000
C_18:1n9t_	3.34 ± 0.16 a	3.39 ± 0.36 a	0.273	3.59 ± 0.25 a	3.20 ± 0.15 b	0.000
C_18:2n6c_	19.43 ± 2.21 a	19.22 ± 1.29 a	0.327	20.94 ± 1.67 a	18.11 ± 0.48 b	0.000
C_18:3n3_	0.68 ± 0.08 a	0.67 ± 0.04 a	0.328	0.73 ± 0.06 a	0.63 ± 0.02 b	0.000
C_18:3n6_	0.19 ± 0.02 a	0.18 ± 0.03 a	0.412	0.17 ± 0.01 b	0.20 ± 0.03 a	0.001
C_20:0_	0.43 ± 0.04 a	0.28 ± 0.12 b	0.000	0.38 ± 0.02 a	0.34 ± 0.16 a	0.310
C_20:1_	0.65 ± 0.11 a	0.37 ± 0.16 b	0.000	0.51 ± 0.02 a	0.51 ± 0.26 a	0.898
C_20:2_	0.64 ± 0.11 a	0.36 ± 0.16 b	0.000	0.50 ± 0.03 a	0.51 ± 0.26 a	0.719
C_20:3n3_	0.75 ± 0.02 a	0.61 ± 0.16 b	0.003	0.72 ± 0.04 a	0.66 ± 0.17 a	0.147
C_20:3n6_	0.53 ± 0.06 a	0.43 ± 0.09 b	0.001	0.49 ± 0.04 a	0.48 ± 0.12 a	0.598
C_20:4n6_	4.24 ± 0.33 b	4.49 ± 0.20 a	0.005	4.51 ± 0.18 a	4.25 ± 0.32 b	0.000
C_20:5n3_	0.66 ± 0.14 a	0.55 ± 0.20 b	0.001	0.79 ± 0.06 a	0.46 ± 0.10 b	0.000
C_22:0_	0.23 ± 0.04 a	0.16 ± 0.05 b	0.000	0.18 ± 0.02 a	0.20 ± 0.08 a	0.180
C_22:6n3_	0.88 ± 0.23 a	0.81 ± 0.24 b	0.027	1.11 ± 0.08 a	0.65 ± 0.09 b	0.000
C_24:0_	0.09 ± 0.01 a	0.07 ± 0.02 b	0.032	0.07 ± 0.01 b	0.09 ± 0.02 a	0.002
C_24:1_	0.14 ± 0.052 a	0.13 ± 0.05 a	0.938	0.08 ± 0.01 b	0.17 ± 0.02 a	0.000

$\bar{x}$—arithmetic mean; s—standard deviation; a, b—means in rows with different letters for sex and hunting districts differ significantly (*p* < 0.05).

**Table 2 animals-12-02394-t002:** Fatty acid sums and indices for the breast muscles of mallard ducks (*Anas platyrhynchos* L.).

Items	Sex		*p*-Value	Hunting District		*p*-Value
	Female (n = 14) $\bar{x}$ ± s	Male (n = 14) $\bar{x}$ ± s		Siedlce (n = 12) $\bar{x}$ ± s	Leszno (n = 16) $\bar{x}$ ± s	
SFA (%) ^1^	30.92 ± 0.10 b	31.79 ± 1.06 a	0.001	30.93 ± 0.27 b	31.68 ± 1.01 a	0.001
MUFA (%) ^2^	41.08 ± 2.90 a	40.81 ± 1.44 a	0.937	39.04 ± 2.11 b	42.37 ± 1.05 a	0.000
PUFA (%) ^3^	28.00 ± 2.81 a	27.33 ± 1.99 a	0.118	29.94 ± 2.10 a	25.95 ± 0.67 b	0.000
n-6 PUFA (%) ^4^	24.38 ± 2.47 a	24.33 ± 1.39 a	0.523	26.11 ± 1.80 a	23.04 ± 0.70 b	0.000
n-3 PUFA (%) ^5^	2.97 ± 0.44 a	2.64 ± 0.60 b	0.005	3.34 ± 0.24 a	2.40 ± 0.34 b	0.000
n-6/n-3 PUFA	8.26 ± 0.40 b	9.66 ± 2.03 a	0.003	7.82 ± 0.03 b	9.82 ± 1.69 a	0.000
PUFA/SFA	0.91 ± 0.09 a	0.86 ± 0.08 b	0.018	0.97 ± 0.07 a	0.82 ± 0.04 b	0.0001
h/H ^6^	3.56 ± 0.30 a	3.44 ± 0.30 b	0.009	3.81 ± 0.16 a	3.26 ± 0.10 b	0.000
AI ^7^	0.27 ± 0.02 b	0.28 ± 0.02 a	0.018	0.26 ± 0.02 b	0.29 ± 0.01 a	0.000
TI ^8^	0.83 ± 0.01 b	0.88 ± 0.06 a	0.000	0.82 ± 0.01 b	0.87 ± 0.05 a	0.000

^1^ SFA (C12:0 + C14:0 + C15:0 + C16:0 + C17:0 + C18:0 + C20:0 + C22:0 + C24:0); ^2^ MUFA (C14:1 + C15:1 + C16:1 + C17:1 + C18:1 n9c + C18:1 n9t + C20:1 + C24:1); ^3^ PUFA (C18:2n6c + C18:3n3 + C18:3n6 + C20:2 + C20:3n3 + C20:3n6 + C20:4n6 + C20:5n3 + C22:6n3); ^4^ n-6 PUFA (C18:2n6c + C18:3n6 + C20:3n6 + C20:4n6); ^5^ n-3 PUFA (C18:3n3 + C20:3n3 + C20:5n3 + C22:6n3); ^6^ h/H (Hypocholesterolaemic/Hypercholesterolaemic index) = (C18:1n9t + C18:1n9c + C18:2n6c + C18:3n3 + C20:3n6 + C20:4n6)/(C14:0 + C16:0); ^7^ AI (atherogenic index) = (4 × C14:0 + C16:0)/(100 − SFA); ^8^ TI (thrombogenic index) = (C14:0 + C16:0 + C18:0)/(0.5 × MUFA + 0.5 × Σ PUFA + 3 × C18:3n3 + 1/(Σn6:n3 PUFA); a, b—means in rows with different letters for sex and hunting districts differ significantly (*p* < 0.05).

**Table 3 animals-12-02394-t003:** Fatty acid profile of the leg muscles of mallard ducks (*Anas platyrhynchos* L.) (%).

Items	Sex		*p*-Value	Hunting District		*p*-Value
	Female (n = 14) $\bar{x}$ ± s	Male (n = 14) $\bar{x}$ ± s		Siedlce (n = 12) $\bar{x}$ ± s	Leszno (n = 16) $\bar{x}$ ± s	
C_12:0_	0.19 ± 0.11 a	0.19 ± 0.12 a	0.809	0.32 ± 0.04 a	0.09 ± 0.02 b	0.000
C_14:0_	0.96 ± 0.08 b	1.00 ± 0.03 a	0.001	1.03 ± 0.04 a	0.95 ± 0.06 b	0.000
C_14:1_	0.42 ± 0.04 a	0.39 ± 0.03 b	0.032	0.40 ± 0.04 a	0.41 ± 0.03 a	0.290
C_15:0_	0.39 ± 0.10 a	0.43 ± 0.05 a	0.170	0.47 ± 0.07 a	0.37 ± 0.06 b	0.000
C_15:1_	0.91 ± 0.48 a	0.76 ± 0.30 b	0.041	0.42 ± 0.14 b	1.15 ± 0.21 a	0.000
C_16:0_	16.79 ± 0.48 a	16.48 ± 0.50 b	0.023	16.20 ± 0.43 b	16.96 ± 0.27 a	0.000
C_16:1_	5.37 ± 0.13 a	5.22 ± 0.20 b	0.030	5.34 ± 0.19 a	5.27 ± 0.18 a	0.146
C_17:0_	0.67 ± 0.16 a	0.73 ± 0.14 a	0.171	0.84 ± 0.13 a	0.60 ± 0.07 b	0.000
C_17:1_	0.36 ± 0.16 a	0.37 ± 0.14 a	0.850	0.30 ± 0.22 a	0.41 ± 0.04 a	0.069
C_18:0_	10.73 ± 0.21 b	11.29 ± 0.24 a	0.000	10.90 ± 0.27 b	11.09 ± 0.39 a	0.013
C_18:1n9c_	35.35 ± 0.96 a	35.53 ± 0.67 a	0.562	35.19 ± 0.92 a	35.63 ± 0.69 a	0.191
C_18:1n9t_	2.91 ± 0.19 a	2.94 ± 0.35 a	0.332	3.18 ± 0.25 a	2.73 ± 0.06 b	0.000
C_18:2n6c_	16.27 ± 0.81 a	16.23 ± 0.72 a	0.306	17.12 ± 0.19 a	15.60 ± 0.08 b	0.000
C_18:3n3_	0.57 ± 0.03 a	0.57 ± 0.02 a	0.307	0.60 ± 0.01 a	0.55 ± 0.01 b	0.000
C_18:3n6_	0.26 ± 0.07 a	0.26 ± 0.07 a	0.768	0.21 ± 0.04 b	0.30 ± 0.06 a	0.000
C_20:0_	0.67 ± 0.14 a	0.66 ± 0.20 a	0.512	0.53 ± 0.14 b	0.77 ± 0.10a	0.000
C_20:1_	1.01 ± 0.12 a	1.01 ± 0.24 a	0.503	0.88 ± 0.20 b	1.11 ± 0.11 a	0.000
C_20:2_	1.00 ± 0.12 a	1.02 ± 0.26 a	0.854	0.87 ± 0.19 b	1.12 ± 0.12 a	0.000
C_20:3n3_	0.46 ± 0.08 a	0.41 ± 0.04 b	0.013	0.39 ± 0.04 b	0.47 ± 0.06 a	0.000
C_20:3n6_	0.75 ± 0.07 a	0.75 ± 0.08 a	0.880	0.71 ± 0.06 b	0.78 ± 0.06 a	0.002
C_20:4n6_	2.38 ± 0.20 a	2.26 ± 0.44 a	0.503	2.51 ± 0.37 a	2.18 ± 0.25 b	0.001
C_20:5n3_	0.35 ± 0.02 a	0.31 ± 0.04 b	0.009	0.35 ± 0.03 a	0.32 ± 0.04 b	0.001
C_22:0_	0.24 ± 0.07 a	0.24 ± 0.08 a	0.978	0.18 ± 0.05 b	0.28 ± 0.06 a	0.000
C_22:6n3_	0.72 ± 0.17 a	0.66 ± 0.25 a	0.145	0.91 ± 0.10 a	0.52 ± 0.10 b	0.000
C_24:0_	0.12 ± 0.04 a	0.12 ± 0.04 a	0.858	0.08 ± 0.02 b	0.15 ± 0.03 a	0.000
C_24:1_	0.15 ± 0.05 a	0.15 ± 0.06 a	0.784	0.09 ± 0.02 b	0.19 ± 0.03 a	0.000

a, b—means in rows with different letters for sex and hunting districts differ significantly (*p* < 0.05).

**Table 4 animals-12-02394-t004:** Fatty acid sums and indices in the leg muscles of mallard ducks (*Anas platyrhynchos* L.) (%).

Items	Sex		*p*-Value	Hunting District		*p*-Value
	Female (n = 14) $\bar{x}$ ± s	Male (n = 14) $\bar{x}$ ± s		Siedlce (n = 12) $\bar{x}$ ± s	Leszno (n = 16) $\bar{x}$ ± s	
SFA (%)	30.74 ± 0.30 b	31.15 ± 0.67 a	0.004	30.53 ± 0.36 b	31.26 ± 0.46 a	0.000
MUFA (%)	46.49 ± 0.90 a	46.38 ± 0.45 a	0.753	45.80 ± 0.51 b	46.91 ± 0.42 a	0.000
PUFA (%)	22.76 ± 0.97 a	22.48 ± 1.01 a	0.102	23.67 ± 0.26 a	21.83 ± 0.50 b	0.000
n-6 PUFA (%)	19.66 ± 0.79 a	19.50 ± 0.96 a	0.194	20.55 ± 0.19 a	18.85 ± 0.32 b	0.000
n-3 PUFA (%)	2.10 ± 0.20 a	1.95 ± 0.30 b	0.039	2.25 ± 0.16 a	1.86 ± 0.20 b	0.000
n-6/n-3 PUFA	9.42 ± 0.79 b	10.16 ± 1.00 a	0.026	9.17 ± 0.63 b	10.26 ± 0.92 a	0.000
PUFA/SFA	0.74 ± 0.04 a	0.72 ± 0.05 b	0.017	0.78 ± 0.01 a	0.70 ± 0.02 b	0.000
h/H	3.28 ± 0.12 a	3.34 ± 0.16 a	0.059	3.45 ± 0.13 a	3.21 ± 0.02 b	0.000
AI	0.30 ± 0.01 a	0.30 ± 0.01 a	0.835	0.29 ± 0.01 b	0.30 ± 0.01 a	0.000
TI	0.82 ± 0.02 b	0.83 ± 0.03 a	0.023	0.80 ± 0.01 b	0.84 ± 0.02 a	0.000

a, b—means in rows with different letters for sex and hunting districts differ significantly (*p* < 0.05).

## Data Availability

No new data were created or analysed in this study. Data sharing is not applicable to this article.

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
