# Peer review of "Analysis of the Fatty Acid Profile of the Tissues of Hunted Mallard Ducks (Anas platyrhynchos L.) from Poland"

_animals, 2022, doi:10.3390/ani12182394_

Round 1

Reviewer 1 Report

It's a different study compared to the ones I'm used to reviewing for Animals. The authors prepared an article to demonstrate the occurrence of different polyunsaturated fatty acids (PUFAs) in mallard meat. My main concern is that the article has a primary focus on the composition of meat (fatty acids) rather than animals (mallards). So I strongly think this article would fit better in a more specialized scientific journal such as “Foods” (also from the MDPI). As pointed out in the first sentence of the Abstract: “The objective of the study was to analyze the fatty acid profile of selected tissues of mallard ducks (Anas platyrhynchos L.), in relation to the place where they were obtained and their sex, with regard to the human diet”. 

If the authors decide to proceed with the submission procedure of this study to Animals (with the main focus on the mallards), I think it is necessary to completely rewrite it as per my comments below:

Ethics: it is necessary to include the legal authorization to perform this study as well as more information of the hunting procedure. 

Title: I suggest to remove the information “in relation to environmental conditions and sex”. The number of samples (28) is too low to achieve any more definitive conclusion about this. 

Simple Summary, Abstract and Introduction: the Introduction is quite long focusing only on the fatty acids meat composition. It would be necessary to rewrite it focusing on mallards hunting and consumption (with a deep review about these subjects, including two or more paragraphs). I think it is also important to explain the mallard diet and to highlight the importance of it for the meat composition. On oppose, the detailed description of the meat composition must be reduced for only one or two paragraphs. Simple Summary could include the importance of PUFAs for human consumption. I think this paragraph would fit perfect in it : “Lifestyle changes and an unsuitable diet have led to the rapid development of civilization diseases and sudden deaths. This is clearly evident to contemporary consumers, who look for foods produced with respect for the natural environment, which also have an original flavor and health-promoting properties, such as game meat.”

Materials and Methods: it is necessary to include more detailed information about “Animals and sample collection”, as I previously mentioned in the topic “Ethics”.  Also a map from the region, including the two hunting districts would be very welcome. I also suggest the authors to explain the probable diet of the mallards from the two districts.  

Results: it is necessary to write more concise paragraphs. I would suggest to compare the difference of the feeding in the two regions. In addition, the data could be presented in Figures, such as bar graphs. It would be more illustrative for the readers.

Discussion: as previously mentioned in the second topic, it would be necessary to rewrite it focusing on mallards diet as well as their hunting and consumption. Please review other articles about these subjects and include in this topic. In addition, a paragraph with the limitations of the study would improve this topic.

Conclusion: should be shortened.

All text: please carefully review it to correct the English grammar mistakes and to inform all used abbreviations.  It is also not necessary to present three numbers after the dot in the percentages values (as 17.897%). Please use only one number after the dot. 

Therefore, the manuscript needs a lot more work before it is ready for another peer review.

Author Response

Thank you for your in-depth review and valuable comments that we used to improve our work. All introduced changes are visible in the text and marked in red with Track Changes. We will try to answer the best we can. (The red parts below are point-by-point responses) 

It's a different study compared to the ones I'm used to reviewing for Animals. The authors prepared an article to demonstrate the occurrence of different polyunsaturated fatty acids (PUFAs) in mallard meat. My main concern is that the article has a primary focus on the composition of meat (fatty acids) rather than animals (mallards). So I strongly think this article would fit better in a more specialized scientific journal such as “Foods” (also from the MDPI). As pointed out in the first sentence of the Abstract: “The objective of the study was to analyze the fatty acid profile of selected tissues of mallard ducks (Anas platyrhynchos L.), in relation to the place where they were obtained and their sex, with regard to the human diet”. 

If the authors decide to proceed with the submission procedure of this study to Animals (with the main focus on the mallards), I think it is necessary to completely rewrite it as per my comments below:

Point 1. Ethics: it is necessary to include the legal authorization to perform this study as well as more information of the hunting procedure. 

Response 1: The mallards were obtained by hunters belonging to the Polish Hunting Association, who are authorized to manage populations of game animals [Act of 13 October, 1995, Hunting Law, Journal of Laws 2022, item 1173]. The hunters culled the ducks in accordance with the Annual Hunting Plan for hunting districts drawn up by the leaseholders of the districts, after consulting the commune administrator and the competent agricultural chamber and subject to approval by the competent district manager of the State Forests National Forest Holding in agreement with the Polish Hunting Association. The hunters made the mallard carcasses available for scientific research in compliance with the Hunting Law.

Point 2: Title: I suggest to remove the information “in relation to environmental conditions and sex”. The number of samples (28) is too low to achieve any more definitive conclusion about this. 

Response 2: I suggest leaving the title unchanged, because it indicates the scope of the research.

Point 3: Simple Summary, Abstract and Introduction: the Introduction is quite long focusing only on the fatty acids meat composition. It would be necessary to rewrite it focusing on mallards hunting and consumption (with a deep review about these subjects, including two or more paragraphs). I think it is also important to explain the mallard diet and to highlight the importance of it for the meat composition. On oppose, the detailed description of the meat composition must be reduced for only one or two paragraphs. Simple Summary could include the importance of PUFAs for human consumption. I think this paragraph would fit perfect in it : “Lifestyle changes and an unsuitable diet have led to the rapid development of civilization diseases and sudden deaths. This is clearly evident to contemporary consumers, who look for foods produced with respect for the natural environment, which also have an original flavor and health-promoting properties, such as game meat.”

Response 3: The introduction has been changed. A 2-paragraph line from 48 to 68 has been added and the paragraphs for the detailed description of the meat composition have been shortened.

The Simple Summary has been revised as suggested, line 12 to 15.

Point 4: Materials and Methods: it is necessary to include more detailed information about “Animals and sample collection”, as I previously mentioned in the topic “Ethics”.  Also a map from the region, including the two hunting districts would be very welcome. I also suggest the authors to explain the probable diet of the mallards from the two districts.  

Response 4: In the Material and Methods section we have added a description of the study area and information about sample collection and the diet of mallards. A map of the location of the studied hunting districts has been attached (lines 24 to 61).

Point 5: Results: it is necessary to write more concise paragraphs. I would suggest to compare the difference of the feeding in the two regions. In addition, the data could be presented in Figures, such as bar graphs. It would be more illustrative for the readers.

Response 5: The paper describes only the significant differences between means for the analysed fatty acids. The likely diet of the mallards in the districts is given in the Material and Methods section and is further discussed in the Discussion section.  

The data presented in the table are clear. Here only two values are compared for sex and two for hunting districts. The probabilities are given next to the values.

Point 6: Discussion: as previously mentioned in the second topic, it would be necessary to rewrite it focusing on mallards diet as well as their hunting and consumption. Please review other articles about these subjects and include in this topic. In addition, a paragraph with the limitations of the study would improve this topic.

Response 6: We have added information from current literature about the diet of mallards. The results of the study are compared in relation to the diet of the ducks from the analysed hunting districts. We have also added a paragraph about the limitations of the study.  

Point 7: Conclusion: should be shortened.

Response 7: The conclusion has been shortened and made more general, lines 550 to 559.

Point 8: All text: please carefully review it to correct the English grammar mistakes and to inform all used abbreviations.  It is also not necessary to present three numbers after the dot in the percentages values (as 17.897%). Please use only one number after the dot. 

Response 8: The entire text has been carefully reviewed and English grammatical errors have been corrected and all abbreviations used have been informed. Due to the small differences between some means, in our opinion giving only the first value after the decimal point would make it impossible to show the significant differences between these values.  

Thank you again for a lot of time dedicated to our work and valuable comments.

Reviewer 2 Report

Review to articles:

Analysis of the fatty acid profile of the tissues of mallard ducks  (Anas Platyrhynchos L.) in relation to environmental conditions  and sex

The presented work comprehensively analyzes the fatty acid profile of breast muscle and leg muscle of wild ducks in two different hunting regions of Poland. In your articles, you compared the differences in the fatty acid profile between males and females and also the differences between these indicators between two different hunting locations. It would be appropriate to supplement the data from the comparison of males and females depending on the hunting location. To the extent that the differences between the sexes were significant, it would be appropriate to add the indicators of the differences between the locations by sex.

Line 451-453: The PUFA/SFA ratio values given in the text do not correspond to the values given in Table 4. Please correct.

Author Response

Thank you very much for your valuable comments. All introduced changes are visible in the text and marked in red with Track Changes.

Point 1: The presented work comprehensively analyzes the fatty acid profile of breast muscle and leg muscle of wild ducks in two different hunting regions of Poland. In your articles, you compared the differences in the fatty acid profile between males and females and also the differences between these indicators between two different hunting locations. It would be appropriate to supplement the data from the comparison of males and females depending on the hunting location. To the extent that the differences between the sexes were significant, it would be appropriate to add the indicators of the differences between the locations by sex.

Response 1: The statistical analysis was performed according to a two-factorial model with interaction (hunting district, sex, hunting district x sex) and showed no significant interaction for the analysed fatty acids. For this reason we presented only the main effects, i.e. of the hunting district and sex.

Point 2. Line 451-453: The PUFA/SFA ratio values given in the text do not correspond to the values given in Table 4. Please correct.

Response 2. The values have been corrected to correspond to the data in Table 4 line 531.

Round 2

Reviewer 1 Report

Peer review:

Manuscript " Analysis of the fatty acid profile of the tissues of mallard ducks (Anas Platyrhynchos L.) in relation to environmental conditions and sex” (animals-1885330.R1)

I have already revised the first version of this manuscript. I pointed out that I was concerned that the article was primarily focused on the composition of meat (fatty acids) rather than animals (ducks). I also suggested transferring the manuscript to a more specialized scientific journal, such as “Foods” (also from the MDPI). It seems that the authors chose to publish in Animals. So they made some efforts to adjust it better to the animals, as per my previous suggestions.

In general, the manuscript has been improved. Now the focus is on animals (ducks) and the entire study has been better explained, including the ethical aspects and a map where the animals were hunted in the methodology. However, there are still issues that have not been properly resolved.

First, I am concerned about the title proposed by the authors. As I commented in my first review: “The number of samples is too low to conclude about environmental conditions and sex”. Also, I think this study describes a specific situation observed in Poland. Therefore, it is necessary to describe these aspects also in the title. Just a suggestion: “Analysis of the fatty acid profile of the tissues of hunted mallard ducks (Anas Platyrhynchos L.) from Poland”.

Furthermore, the Results have not been modified as per my previous suggestion. It is really necessary to write more concise and clear paragraphs. It is difficult to understand the Results as described. I also suggested earlier to compare the difference in animal diet in the two regions. It is necessary to demonstrate this comparison also in the Results (and not only in the Discussion). Finally, display only one number after the dot in frequencies (maximum 2!). If the differences are so small that three numbers after the dot are required, the differences are not significant and these analyzes should not be presented in the manuscript. The help of a statistician is strongly recommended! Also one more reason to change the title!

Therefore, the manuscript needs more work before it is ready.

Author Response

Thank you very much for your review and valuable comments, which we have used to improve the manuscript. All changes in the manuscript are shown in red using the Track Changes function.

Point 1. 

First, I am concerned about the title proposed by the authors. As I commented in my first review: “The number of samples is too low to conclude about environmental conditions and sex”. Also, I think this study describes a specific situation observed in Poland. Therefore, it is necessary to describe these aspects also in the title. Just a suggestion: “Analysis of the fatty acid profile of the tissues of hunted mallard ducks (Anas Platyrhynchos L.) from Poland”.

Response 1. 

We have changed the title as suggested.

Point 2. 

Furthermore, the Results have not been modified as per my previous suggestion. It is really necessary to write more concise and clear paragraphs. It is difficult to understand the Results as described. I also suggested earlier to compare the difference in animal diet in the two regions. It is necessary to demonstrate this comparison also in the Results (and not only in the Discussion). Finally, display only one number after the dot in frequencies (maximum 2!). If the differences are so small that three numbers after the dot are required, the differences are not significant and these analyzes should not be presented in the manuscript. The help of a statistician is strongly recommended! Also one more reason to change the title!

Response 2.

We have rewritten the Results section as suggested, with more concise paragraphs. The results are compared with respect to the differences in the diet in the two regions. We have shown only 2 digits after the decimal point.

We have rewritten the Results section as suggested, with more concise paragraphs. The results are compared with respect to the differences in the diet in the two regions. We have shown only 2 digits after the decimal point.

Thank you very much for improving our manuscript.

Round 3

Reviewer 1 Report

The authors improved the article as recommended.